# Fabrication of a 4 m SiC Aspheric Mirror Using an Optimized Strategy of Dividing an Error Map

Zhenyu Liu [1,2], Longxiang Li [1,2,*], Erhui Qi [1,2], Haixiang Hu [1,2] and Xiao Luo [1,2]

1    Key Laboratory of Optical System Advanced Manufacturing Technology, Chinese Academy of Sciences, Changchun 130033, China; liuzhenyu@ciomp.ac.cn (Z.L.); hhx@ciomp.ac.cn (H.H.); luoxiao@ciomp.ac.cn (X.L.)
2    Changchun Institute of Optics, Fine Mechanics and Physics, Chinese Academy of Sciences, Changchun 130033, China
*    Correspondence: lilx@ciomp.ac.cn

**Abstract:** This paper introduces an optimization strategy for fabricating large aspheric mirrors. We polished a large SiC aspheric mirror, 4 m in diameter, achieving a surface error of $1/40\lambda$ RMS. To the best of our knowledge, this is the first instance of such a result for a mirror of this material and size combination. Due to the various performance settings of different tools, achieving optimal polishing results with a single setting is challenging. We evaluated the performance of various tool settings and developed an optimization strategy, dividing error maps to enhance efficiency in large-aperture aspheric mirror fabrication. We established the relationship between tool size and its error control capability. The residual error map of the mirror was divided into two parts using Zernike polynomial expansion based on the frequency order of the error map. Here, we used the first 36 terms of the Zernike polynomial fit to define a low-order error map, and the residual error was used to define a high-order error map. Large tools were used to correct the low-order frequency error map, whereas small tools were used to correct the high-order frequency error map. Therefore, the original residual error map could be corrected with significantly high efficiency. By employing this strategy, we fabricated a 4 m SiC aspheric mirror in 18 months, achieving a final surface error better than $0.024\lambda$ RMS.

**Keywords:** computer-controlled polishing; silicon carbide mirror; dwell time algorithm; low-frequency error; high-frequency error; error map dividing strategy

## 1. Introduction

Aspherical optical elements are extensively used in modern optical systems due to their ability to enhance system performance. Firstly, these elements correct aberrations more effectively, as they can precisely match the ideal optical wavefront, thereby delivering clearer images. Secondly, aspherical elements offer greater design flexibility, enabling them to replace combinations of spherical mirrors. Consequently, the entire optical system achieves comparable or superior performance with fewer components, simplifying the design and reducing weight.

With advancements in optical technology, next-generation optical systems increasingly demand higher resolution. Meeting this requirement necessitates enlarging the primary mirror's diameter. For instance, the primary mirror of the James Webb Space Telescope (JWST) measures 6.5 m [1]. The situation is similar for ground-based telescopes, such as the Large Binocular Telescope (LBT), which has an 8.4 m primary mirror [2]; the Subaru Telescope, which has an 8.2 m primary mirror [3]; and the Very Large Telescope (VLT), which has an 8.2 m primary mirror [4].

Furthermore, the Giant Magellan Telescope (GMT) has a 24.5 m diameter [5], the Thirty Meter Telescope (TMT) has a 30 m diameter [6], and the European Extremely Large Telescope (EELT) has a 39.3 m diameter [7]. Telescopes are being developed with

significantly larger diameters, reaching tens of meters. ULE® and ZERODUR® materials are widely used in the manufacturing of large-aperture aspherical mirrors [8,9]. Nevertheless, due to its superior mechanical and thermal properties, SiC has been regarded as the preferred material for large lightweight spaceborne mirrors [10]. However, SiC's high specific stiffness presents challenges in grinding and polishing [11].

Regarding mirrors made of SiC material, Reosc and Boostec recently reported the fabrication of a 1.54 by 0.49 m SiC primary mirror, achieving a final processing result of 9 nm RMS [12]. The SiC primary mirror of the EUCLID Telescope, with an aperture of 1.25 m, has a targeting precision of 9 nm [13]. The Herschel Telescope's primary mirror, previously the largest SiC material reflector, attained a surface form accuracy of 3 micrometers RMS [14].

Computer-controlled optical surfacing (CCOS) is an advanced technique that has been widely used in the fabrication of aspheric optics [15–17]. It uses a computer to control the position and movement of a tool, with a diameter much smaller than that of an aspherical workpiece, to grind and polish. The calculated tool dwell-time distribution ensures deterministic material removal. However, with an increase in the diameter of the mirror, the amount of material removal increases significantly, especially for a SiC mirror, which requires a high material removal rate. Thus, improving the material removal rate while ensuring machining accuracy is becoming a great challenge in the machining process of large-aperture aspherical SiC mirrors [17]. Kim et al. proposed a non-sequential technique and successfully applied it to process the GMT's primary mirror [18]. Our previous research studied a combined processing technology for producing large-aperture aspheric mirrors with a proposed multi-mode optimization based on matrix operation [19].

Recently, the Changchun Institute of Optics, Fine Mechanics and Physics (CIOMP) completed the manufacturing of the world's largest 4 m aperture silicon carbide (SiC) material mirror. Xianggang Luo introduced the significance of this work [20]. Additionally, Xuejun Zhang provided a comprehensive overview of the entire manufacturing process of this mirror, including blank preparation technology, processing technology, and high-precision detection technology [21]. This paper offers a more detailed introduction to the processing technology used in the manufacturing process of this mirror. It primarily focuses on introducing efficient processing strategies for ultra-large-aperture aspherical SiC mirrors.

In this research, the relationship between the tool sizes, material removal rate, and surface error convergence efficiency was studied. An optimized strategy based on error dividing was established.

We decomposed the surface error map into low- and high-order components based on Zernike polynomials. Zernike polynomials are an effective tool for analyzing error maps [22,23], and we chose this method for the following reasons:

- Zernike polynomials can provide a smooth error map fit without high-frequency oscillation.
- Zernike polynomials are orthogonal and do not contribute to the rest of the error.
- Zernike polynomials can characterize the errors caused by the design tolerance. Errors of tolerance can be eliminated by adjusting the coefficient.

Here, we defined the first 36 terms of the Zernike polynomials as low-order components. Then, the low-order errors were corrected using a large tool with a high material removal rate. Next, a small tool was used to correct the residual errors, which were composed of high-order frequency errors and residual errors caused by the large tools. Finally, we achieved the efficient removal of surface form errors across the entire frequency. This processing strategy, which combined the advantages of large and small grinding heads, could achieve manufacturing with a high efficiency and precision. This strategy was successfully applied to the fabrication of a 4 m SiC aspheric mirror with high efficiency and good accuracy.

## 2. Principle of Error Map Dividing Strategy

### 2.1. Relationship between Tool Size and Surface Figuring Efficiency

Tool size is one of the key parameters in the optical manufacturing process. D.D. Walker studied the mid-spatial frequencies caused by the misfit between aspheric surfaces and tools [24]. Du H studied the relationship between tool size and mid-to-high-frequency control ability [25]. Both studies described the tool size and its figuring ability well. With regard to surface figuring efficiency, it is essential to consider not only the figuring ability but also the material removal rate. For a given tool, the material removal rate and figuring ability are two important indicators. Generally, the material removal rate increases with an increase in the size of the tool, whereas the figuring ability decreases because large machining tools suffer from poor accuracy. To investigate the relationship between tool size, material removal rate, and figuring ability, we conducted a series of simulation experiments.

First, we generated an error map described by a sine function with different periods, as shown in Equation (1):

$$z = \frac{1}{2}\sin(2 \times \pi \times r/p), \tag{1}$$

Here, $z$ is the error value, $r$ is the radius of the data points, and $p$ is the period of the error map. The diameter of the simulation error map was 1000 mm. The range of the period used in the simulation was from 50 to 1000 mm. One of the error maps with a period of 200 mm is shown in Figure 1.

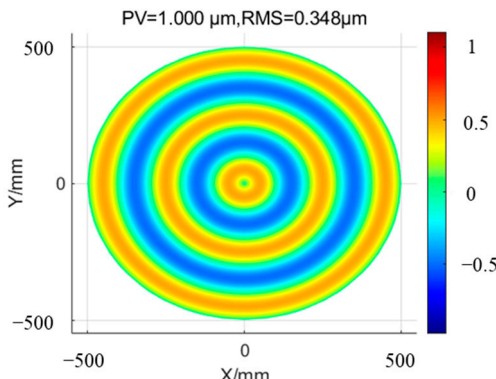

**Figure 1.** Simulation error map with a period of 200 mm.

Second, we used tools with different sizes to calculate the dwell time and to obtain an ideal residual error map. The results are shown in Figure 2.

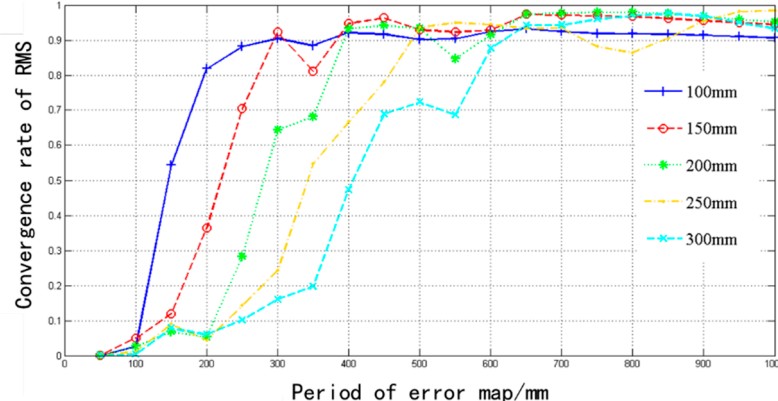

**Figure 2.** Convergence rate as a function of the period of the error map for different tool sizes.

Here, we define the convergence rate as a function of

$$convergency\ rate = \frac{\text{RMS}_{\text{ini}} - \text{RMS}_{\text{simulated}}}{\text{RMS}_{\text{ini}}} \cdot 100\%, \tag{2}$$

$\text{RMS}_{\text{ini}}$ represents the initial surface form error, and $\text{RMS}_{\text{simulated}}$ is the result of the simulation calculation. Figure 2 shows the simulation results of removing the surface shape errors of different periods with different tool sizes. It can be seen from the simulation results that, for a surface shape error with a period greater than 600 mm, the maximum convergence efficiency of >90% was achieved, which was independent of the tool size. With the decrease in the surface error period, the convergence efficiency of the large-diameter tools decreased much more rapidly than that of the small tools. Here, "convergence efficiency" refers to the rate at which surface form errors decrease over time.

To investigate the relationship between the tool size and manufacturing time, we calculated the dwell time as a function of the tool size on the same error map. A surface shape error with a frequency period of 650 mm was selected because of the high convergence for different tool sizes. The results of the virtual simulation are shown in Figure 3. Table 1 shows the corresponding data of the convergence rates and dwell time under different tool sizes.

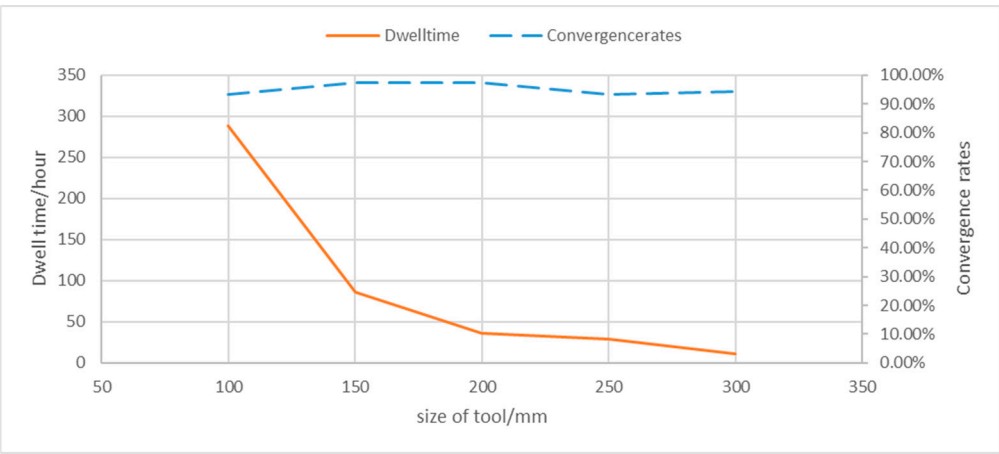

**Figure 3.** Dwell time and convergence rate as a function of tool size.

**Table 1.** Calculated dwell time and convergence rate as a function of tool size.

| Tool Size (mm) | 100 | 150 | 200 | 250 | 300 |
|---|---|---|---|---|---|
| Convergence rates | 93.4% | 97.4% | 97.5% | 93.4% | 94.3% |
| Dwell time (h) | 288.8 | 85.7 | 36.2 | 29.2 | 10.8 |

Figure 3 and Table 1 show similar convergence rates of the error map with a 650 mm frequency. However, the dwell time of a small tool (100 mm) was about 30 times that of a large tool (300 mm). Thus, in this case, using a large tool could significantly improve the processing efficiency.

Real error maps are usually composed of various complex factors, especially for a large-aperture aspherical mirror. Since its error period may cover a range from several millimeters to meters, it is difficult to realize high-efficiency and high-precision machining using a single-sized tool or machining method. Hence, a new strategy to combine different tools with different methods is presented in this paper to realize high-efficiency and high-precision manufacturing.

## 2.2. Principle of the Error Dividing Strategy

The basic principle of the error dividing strategy was to utilize a certain algorithm, such as the 2Sigma formula, the Fast Fourier Transform (FFT) algorithm, or Zernike polynomials,

to divide the error map into different frequencies. A large tool was used to remove low-order frequency errors, and a small tool, MRF (magnetorheological finishing) or IBF (ion beam figuring), was used to remove medium- and high-frequency errors. By using this method, low-order frequency errors could be removed with a large tool more effectively with a high convergence efficiency. Meanwhile, a small tool focused on medium- and high-frequency error control with less material removal but high precision. Therefore, by handling an error map with different tools and methods based on its frequency, the processing strategy that we developed here could obtain a high material removal rate with a high convergence efficiency for producing high-quality large aspheric mirrors. Figure 4 shows a flowchart of this error dividing strategy.

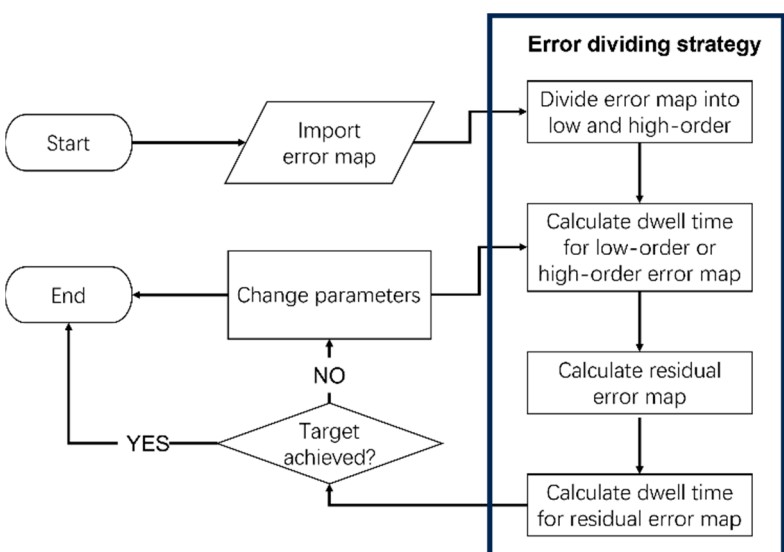

**Figure 4.** Flowchart of error dividing strategy.

First, we manufactured tools of various sizes according to the dimensions of the mirror. Taking the 4 m SiC mirror mentioned in this paper as an example, the sizes of the tools that we made included diameters of 600, 400, and 100 mm, and MRF was used in later stages. Then, we proceeded with the optimization process, as shown in the flowchart. If the simulation results do not meet the requirements, it is necessary to modify the parameters, which primarily refer to the combinations of different-sized tools. The optimization process went through numerous iterations and ultimately satisfied the processing requirements.

### 2.3. Mathematical Model of the Strategy

The error map of the optical mirror surface is characterized by a series of discrete data points. For the convenience of calculation, the surface error can be described in a matrix with an error at each point $(x, y)$ defined as follows:

$$Error_i = [x_i, y_i, z_i], \tag{3}$$

where $i$ represents the $i_{th}$ point at the position $(x_i, y_i)$ on the workpiece, and $z_i$ represents the residual error at this point.

Several methods can be used to divide an error map into low-order errors and middle–high-order errors, including the average method, the 2Sigma formula, and the Fast Fourier Transform (FFT) algorithm. The use of Zernike polynomials is an effective method to fit the surface shape of an optical aspheric mirror [26]. The advantage of using Zernike polynomials for fitting low-order surface errors lies in its capability to effectively eliminate errors induced by misalignment.

Hence, it was adopted in this work to fit the low-order part of the surface shape error using the following mathematical expression:

$$Z_n^m(\rho, \varphi) = \begin{cases} R_n^m(\rho)\cos(m\varphi), m > 0; \\ R_n^{-m}(\rho)\sin(-m\varphi), m < 0; \\ R_n^0(\rho), m = 0. \end{cases} \tag{4}$$

Here, we used the "fringe" indexing scheme, which starts at 0 instead of 1 (subtract 1) [27]. $Z$ is the total value of the polynomial. $\rho$ is the radius of the data points, whereas $\varphi$ is the angle of the data points. $n$ is the radial order, and m is the annular frequency. Depending on the polarity of the m value, $Z$ is divided into three formulas, corresponding to $m > 0$, $m < 0$, and $m = 0$. The coefficient $R$ is represented as follows:

$$R_n^m(\rho) = \sum_{k=0}^{\frac{n-m}{2}} \frac{(-1)^k(n-k)!}{k!\left(\frac{n+m}{2}-k\right)!\left(\frac{n-m}{2}-k\right)!}\rho^{n-2k}. \tag{5}$$

In this study, we used the first 36 terms of the Zernike polynomials to fit the low-order frequency error map. The implementation process is as follows:

(1)    The error map is changed from the Cartesian coordinate system ($x_i$, $y_i$, $z_i$) to the polar coordinate system ($\rho_i$, $\varphi_i$, $z_i$).

(2)    By using Zernike polynomials, error $z_i$ can be expressed as

$$z_i = k_1\rho_i\cos\varphi_i + k_2\rho_i\sin\varphi_i + k_3(2\rho_i^2 - 1) + \dots. \tag{6}$$

where $k_i$ is the coefficient of the $i$th terms in the polynomial. Based on the errors at all the points, we can obtain the least square solution of the $k_i$ fitting.

(3)    The first 36 terms of the polynomials are chosen to describe the error map, which forms the low-order frequency of the error, $Z_{low}$.

*2.4. Dwell Time Optimization Process*

In a CCOS process, the dwell time distribution is determined by the desired amount of material removal of the workpiece; this is a discrete two-dimensional (2D) convolution operation of the dwell-time function and the removal function, which is represented by the symbol "**" in Equation (7). Before grinding or polishing, the surface error map, *E(x,y)*, represents the desired amount of material removal on the optical surface. *R(x,y)* is the tool removal function or influence function, which is closely related to the tool's parameters. *D(x,y)* is the dwell time that should be solved. The value of *D(x,y)* is non-negative or positive. According to Equation (7), the process to obtain the dwell time is a discrete 2D deconvolution in algebra [15,18].

$$E(x,y) = R(x,y) * *D(x,y). \tag{7}$$

Since we used a series of discrete data points in the error map, a matrix-based algorithm can be used to calculate the optimal dwell time [27], as shown in Equation (8):

$$\begin{bmatrix} e_1 \\ \vdots \\ e_m \end{bmatrix} = \begin{bmatrix} r_{11} & \cdots & r_{1n} \\ \vdots & \ddots & \vdots \\ r_{m1} & \cdots & r_{mn} \end{bmatrix} \begin{bmatrix} d_1 \\ \vdots \\ d_n \end{bmatrix}. \tag{8}$$

$E$ is the vector form of the error map, $R$ is the tool removal matrix, and $D$ is the dwell time vector. With the generated tool removal matrix $R_{large}$ for a large-sized tool and the low-order part of the error function $Z_{low}$ from Equation (6), the low-frequency part's dwell

time $D_{low}$. can be calculated from the deconvolution of Equation (9) [28,29]. The material removal $Z_{low\_removal}$ based on the optimal solution can be expressed as

$$Z_{low\_removal} = R_{\text{large}} \times D_{low}. \tag{9}$$

Here, $R_{large}$ is the large tool removal matrix, and $D_{low}$ is the low-order error map's dwell time vector that we want to calculate. There is no general solution to the dwell time in Equation (9). This means that $Z_{low\_removal}$ is not equal to $Z_{low}$. If the initial measured error map is $Z_{ini}$, the residual error, $Z_{residual}$, after removal, $Z_{low\_removal}$, can be calculated as

$$Z_{residual} = Z_{ini} - Z_{low\_removal}, \tag{10}$$

By taking the high-order frequency surface error map $Z_{residual}$ as the target and the removal functions of a small tool, the magnetorheological tool, and the ion beam as the small tool removal matrix $R_{small}$, the dwell time $D_{high}$ of the small tool can be calculated.

$$Z_{residual} = R_{small} \times D_{small}, \tag{11}$$

Through the above method, the residence time distributions of large and small tools after surface error separation can be obtained. The combined strategy established in this way can achieve complementary advantages with improved efficiency without sacrificing accuracy.

### 3. Simulation and Experiment

To verify the effectiveness of the optimized strategy, it was tested in the polishing of a 4 m SiC parabolic mirror. As shown in Figure 5, a computer-controlled FSGJ-4500 polishing machine (designed and manufactured by Changchun Institute of Optics, Fine Mechanics and Physics) with a capacity of 4.5 m was used to produce the 4 m SiC parabolic mirror.

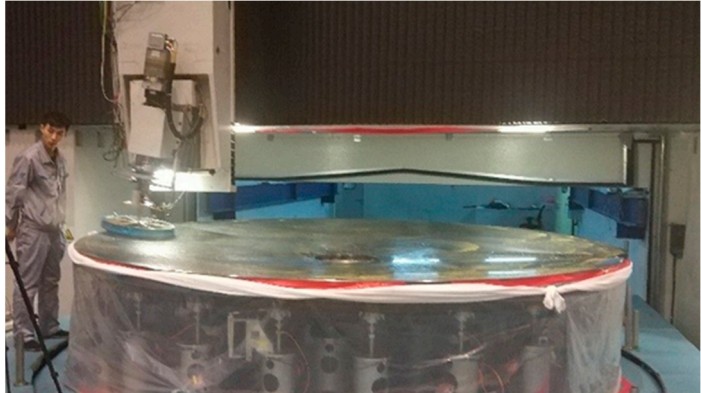

**Figure 5.** The 4 m SiC mirror and FSGJ-4500.

The Tool Influence Function (TIF) is the instantaneous amount of material removal by a tool executing a specific motion. The TIF can be determined either by using empirical measurements or simulations based on mathematical models. In practical applications, our tool operates with a planar rotary motion. Therefore, we employed a mathematical model of the planar rotary removal function to simulate and generate the TIF for computational purposes [30]. The simulation results of the TIF for tools with diameters of 100 and 400 mm are presented as Figure 6.

The surface error map used in the experiment is shown in Figure 7a, and it was divided into a low-order frequency error map and a high-order frequency error, as shown in Figure 7b,c.

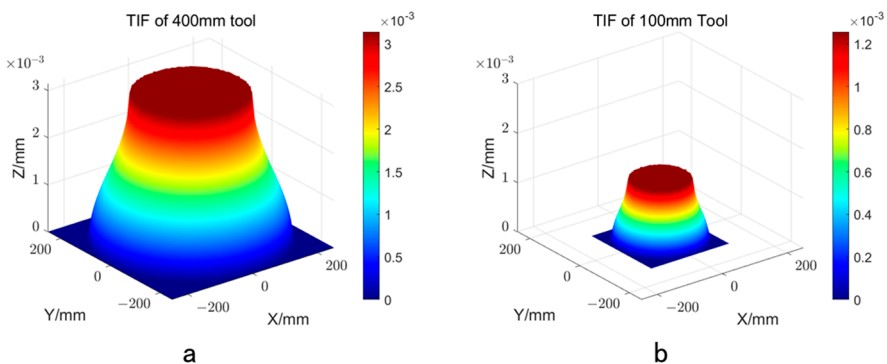

**Figure 6.** Tool removal function: (**a**) 400 mm tool and (**b**) 100 mm tool.

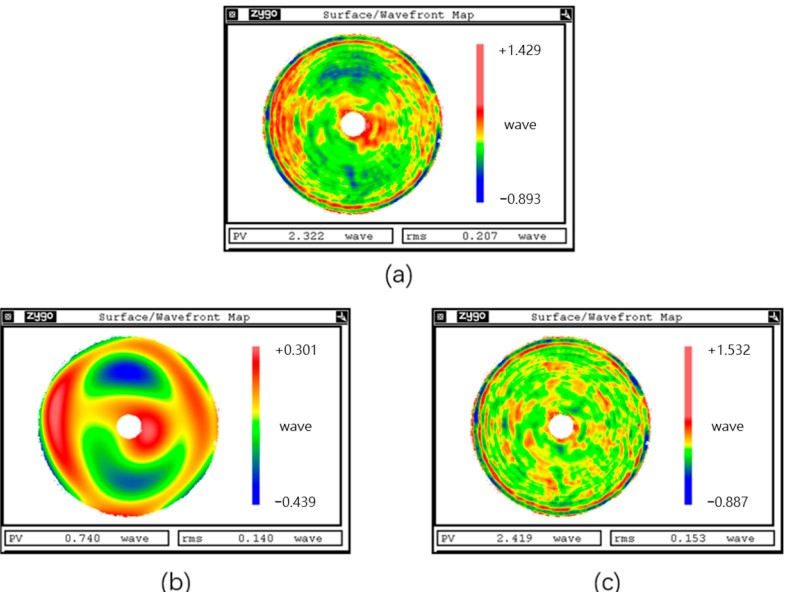

**Figure 7.** Surface error maps: (**a**) initial surface error map, (**b**) low-order error map, and (**c**) high-order error map.

The dwell time vectors were calculated for the low-order and high-order error maps. With the input error data from the low-order error map in Figure 7b, the low-order error map is shown in Figure 8a, and simulated material removal map is shown in Figure 8b. The residual error map is displayed in Figure 8c, with the total simulation result shown in Figure 8d. The simulation results showed that the surface figure accuracy converged from RMS0.2$\lambda$ to RMS0.071$\lambda$ after one optimized iteration.

To verify the effectiveness of the optimized strategy, we also carried out simulation calculations for the large and small tools. The simulation results are displayed in Figure 9 and summarized in Table 2.

**Table 2.** Simulation results.

| Tool Size (mm) | 100 | 400 | 400 and 100 |
|---|---|---|---|
| Initial RMS ($\lambda$) | | 0.207 | |
| Residual RMS ($\lambda$) | 0.057 | 0.179 | 0.071 |
| Convergence rate | 72.46% | 13.53% | 65.70% |
| Time (hour) | 100.6 | 26.6 | 24.1(small) + 11.7(large) = 35.8 |
| Convergence efficiency | 0.72 | 0.51 | 1.84 |

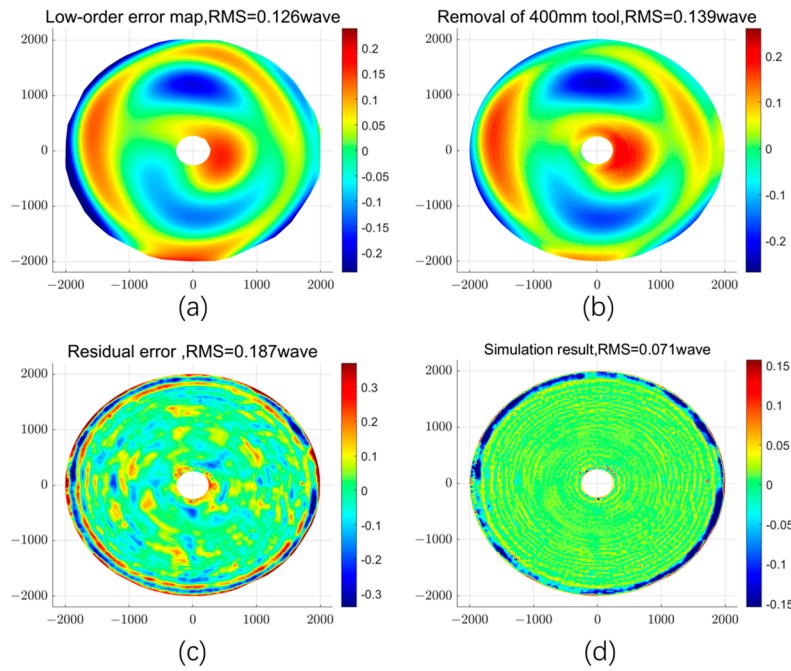

**Figure 8.** Simulation results: (**a**) residual error map for low-order error map, (**b**) calculated material removal for low-order error map, (**c**) residual error map for 100 mm tool figuring, and (**d**) total simulation result.

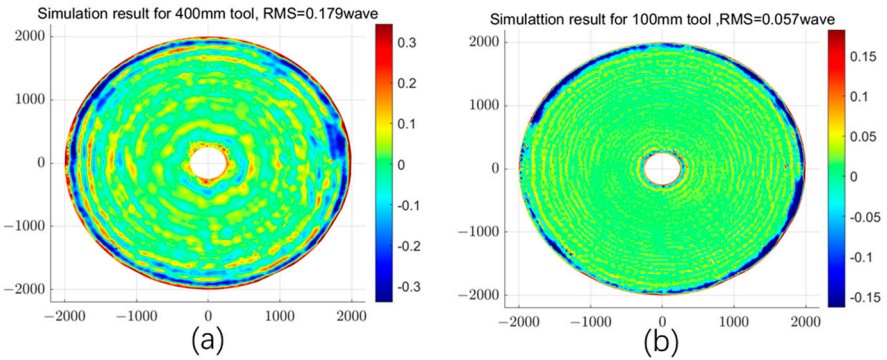

**Figure 9.** Simulation results for large and small tools. (**a**) Simulation result for 400 mm tool. (**b**) Simulation result for 100 mm tool.

To evaluate the effectiveness of the strategy, we defined a parameter of CE (convergence efficiency, percent per hour), which can be calculated as follows:

$$\text{CE} = \frac{Convergence\ rate}{Time} \qquad (12)$$

CE characterizes the convergence rate of the surface error per unit processing time. As shown in Table 2, using the error map dividing strategy could significantly improve the processing efficiency. When a single tool was used, the CE values were 0.72 and 0.51 with tool sizes of 100 and 400 mm, respectively. With the dividing strategy, allowing the 400 mm tool to remove low-order frequency errors, followed by using the 100 mm tool to remove high-order frequency errors, the combined CE increased to 1.84.

Finally, the optimized strategy was tested in an actual fabrication process. Figure 10 shows the surface errors after two tool fabrication processes.

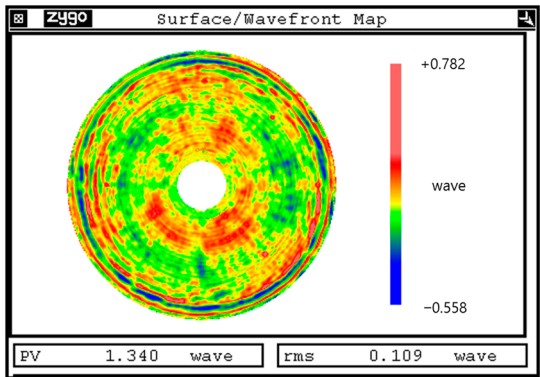

**Figure 10.** Surface errors after fabrication.

After this round of processing, the RMS of the surface error was 0.109λ, with an actual convergence rate of 47%.

The total processing time from grinding to polishing was 18 months. This duration included not only the time directly spent polishing but also encompassed preparatory and intermittent processes such as setup, inspection, and adjustments based on ongoing results.

The total time spent on processing and measurement activities was 377 days, with the measurement phase consuming over half of this duration. A significant amount of time devoted to measurement can be attributed to the challenges associated with obtaining stable and trustworthy measurement data for high-precision, large-aperture (4 m class) aspheric surfaces. The pure machining time was approximately 180 days. The simulations predicted a required polishing time of 120 days, which was based on factors like material removal rate and surface error distribution. The difference between the simulated and actual "pure" machining times could be attributed to unforeseen material behavior, tool wear rates, and edge effect control.

We also conducted simulation calculations for the individual use of 100 mm small tools and 400 mm large tools. The processing precision achieved using the 400 mm large tool alone could not meet the requirements. In contrast, when using the 100 mm small tool alone, the simulated calculation for the required pure machining time was approximately 300 days, significantly higher than when employing an error separation processing strategy.

By employing this strategy for the processing of a 4 m SiC mirror, the final RMS of the error map was 0.024λ, as shown in Figure 11.

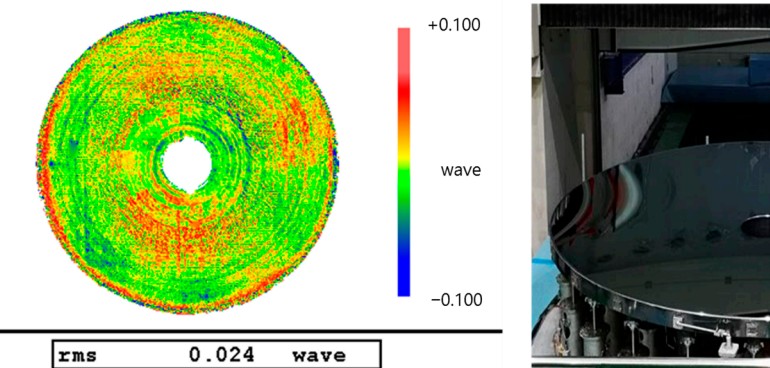

**Figure 11.** Final result of 4 m SiC mirror.

## 4. Discussion

Based on the principle of dividing errors, we present an efficient optimized strategy for processing large-sized aspheric mirrors. The surface figure errors were divided into low-order and high-order frequency errors. Different processing methods were selected according to the characteristics of the errors. A targeted processing strategy was established

based on a mathematical simulation. This strategy overcame the limitation of processing methods using a single tool and was especially suitable for the processing of large-aperture aspherical mirrors, which require the use of a series of tools. The strategy was applied to the polishing of a 4 m SiC mirror. The initial error map RMS was 0.207λ, which converged to 0.109λ after one iterative process. The results of the simulation and experiment demonstrated the effectiveness of the optimized error dividing strategy. Under the guidance of this strategy, the final residual surface error of the 4 m SiC mirror was better than 0.024λ RMS.

However, during the experiments and actual use, we also identified the shortcomings of this method. Firstly, there was a discrepancy between the simulation results and actual processing outcomes, primarily due to the instability in the CCOS processing technique, necessitating further improvements in the certainty of the processing technology. Secondly, the current relationship between the grinding head size and frequency error is still based on simulation results. A series of experiments are needed to determine the relationship between different tool sizes and frequency errors in order to further refine our model and achieve better results.

**Author Contributions:** Conceptualization, Z.L. and L.L.; methodology, Z.L.; software, Z.L.; validation, X.L. and H.H.; formal analysis, Z.L.; investigation, L.L.; resources, Z.L.; data curation, Z.L.; writing—original draft preparation, Z.L.; writing—review and editing, Z.L.; visualization, E.Q.; supervision, X.L.; project administration, X.L.; funding acquisition, L.L. All authors have read and agreed to the published version of the manuscript.

**Funding:** This research was funded by the Youth Innovation Promotion Association CAS (2021215); the National Natural Science Foundation of China (62275246); and the National Key Research and Development Program (2022YFB3403405).

**Institutional Review Board Statement:** Not applicable.

**Informed Consent Statement:** Not applicable.

**Data Availability Statement:** The data presented in this study are available on request from the corresponding author.

**Conflicts of Interest:** The authors declare no conflict of interest.

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
