# Peer review of "Fabrication of a 4 m SiC Aspheric Mirror Using an Optimized Strategy of Dividing an Error Map"

_photonics, doi:10.3390/photonics11020125_

Round 1

Reviewer 1 Report

Comments and Suggestions for Authors

The author proposed a strategy based on error map dividing for fabrication of 4m SiC aspheric mirror. From the paper, the suggested method can improve the fabrication efficiency in both convergence rates and reducing processing time. Below are my specific comments and suggestions.

1.       How many cycles of error map dividing strategy has been applied?

2.       Figure 4 shows the flow chart for error dividing strategy, it says if the target RMS is not achieved, the parameters will be changed. What are those parameters?

3.       The font size is not consistent throughout the whole paper.

4.       The font size inside some figs are too small.

5.       It would be better to provide a photo of the polished mirror.

Comments on the Quality of English Language

The English presentation quality of the entire article needs to be strengthened.

Author Response

Dear Reviewer,

Subject: Manuscript ID photonics-2757637, titled "Fabrication of a 4 m SiC aspheric mirror by an optimized strategy of dividing an error map"

We would like to express our sincere gratitude to you and the reviewers for the time and effort spent reviewing our manuscript. We have carefully considered the comments and suggestions provided and have made corresponding revisions to our manuscript. Below, we address each comment in detail and revisions to the manuscript had been highlighted:

  1. Comment: How many cycles of error map dividing strategy has been applied?

Response: The mirror has undergone 154 rounds of processing, of which 103 rounds were carried out using this strategy. The reason for not fully utilizing this strategy from the beginning was that the initial surface shape errors were primarily of low order, and this strategy was not employed at that stage.

  1. Comment: Figure 4 shows the flow chart for error dividing strategy, it says if the target RMS is not achieved, the parameters will be changed. What are those parameters?

Response: The main parameter modified was the combination of Tools of different sizes, and the related description has already been revised on page 6 of the article.

  1. Comment: The font size is not consistent throughout the whole paper.

Response: We have thoroughly reviewed the entire document and have now standardized the font size to ensure uniformity throughout the paper. These corrections have been made to enhance the readability and professional presentation of our work.

  1. Comment:    The font size inside some figs are too small.

Response: To address this issue, we have revised these figures, ensuring that the font size is larger and consistent with the readability standards for our paper. We have also ensured that this revision does not compromise the clarity and presentation of the data.

  1. Comment: It would be better to provide a photo of the polished mirror.

Response: we have updated Figure 11 with a new photograph of the polished mirror. In this revised figure, we have taken special care to ensure that the font size is adequately large for clear readability without compromising the quality and detail of the image.

We believe these revisions have significantly improved the manuscript and hope that it now meets the journal's standards for publication. We appreciate the opportunity to refine our work and thank you for your contributions to this process.

Zhenyu Liu

Key Laboratory of Optical System Advanced Manufacturing Technology, Chinese Academy of Sciences

[email protected]

Reviewer 2 Report

Comments and Suggestions for Authors

See the attachment 

Comments on the Quality of English Language

The first paragraph of the introduction should be revised to improve the English.

Some dictation problems must be addressed, not limited to “previ-ous” in line 15, “os-cil-lation” in Line 25, “en-suring” in line 12, “inFig-ure3” in Line 72, and so on. 

In Line 51: the word of “following” should be removed. In line 129, “zi” should be replaced by zi (z sub i).

 Line 23; the reasons you provided should be presented as bullet points. The font in Line 37 should be corrected. 

Line 171; Figure 6 should cited instead of “Error! Reference source…..”

Author Response

Dear Reviewer,

Subject: Manuscript ID photonics-2757637, titled "Fabrication of a 4 m SiC aspheric mirror by an optimized strategy of dividing an error map"

We would like to express our sincere gratitude to you and the reviewers for the time and effort spent reviewing our manuscript. We have carefully considered the comments and suggestions provided and have made corresponding revisions to our manuscript. Below, we address each comment in detail and revisions to the manuscript had been highlighted:

  1. Comment: Some of the coauthors of this paper published two other papers, with other coauthors claiming that they fabricated the world's largest SiC aspheric mirror. Those two papers, which are about the same large SiC mirror, are entitled "High-precision fabrication of 4m SIC aspheric mirror" and "Challenges and strategies in high-accuracy manufacturing of the world’s largest SiC aspheric mirror". Both papers have been published in Light Science & application journal, one in Jan 2023 and the second one in Oct 2022. Those papers were not explained in this paper at all and were not used even as references either. It is not clear who actually fabricated the world's largest SiC mirror. Since those two papers and this submission are about the fabrication of the same mirror, the author must clearly and appropriately cite those two papers and clearly explain the goals, objectives, and differences of those papers, at least in one paragraph in their introduction section.

Response: We acknowledge the importance of these publications, as they are indeed closely related to our current work and are authored by some of the co-authors of this paper. The lack of reference to these significant works was an oversight on our part, and we apologize for any confusion it may have caused regarding the actual fabrication of the SiC mirror.

To address this issue, we have revised our manuscript to include a detailed paragraph in the introduction section. This paragraph now clearly cites both papers and elaborately explains the goals, objectives, and differences between these studies and our current submission. Our intention is to provide a comprehensive context and to clarify the unique contributions and perspectives of each paper, ensuring a clear understanding of the collective efforts and advancements in fabricating the world's largest SiC aspheric mirror.

We believe that these additions and clarifications will significantly enhance the understanding of our paper's position within the broader context of SiC mirror fabrication research.

  1. Comment: The first paragraph of the introduction should be revised to improve the English. What do you mean by “with a simplified structure”? Why are the sizes of the three mirrors put in parentheses?

Response: The introductory paragraph has been thoroughly revised in response to the reviewer's comments. To address the query regarding 'a simplified structure,' the revised text now clearly explains that incorporating aspherical optical elements significantly increases design flexibility. This, in turn, enables the substitution of multiple spherical optical components with fewer aspherical ones, effectively simplifying the overall system structure. Regarding the sizes of the three mirrors being originally presented in parentheses, we acknowledge that this formatting choice may have led to some ambiguity. To resolve this, their dimensions have been seamlessly integrated into the main body of the text. This revision not only smoothens the narrative flow but also more directly underscores the relevance of these dimensions to our study.

  1. Comment: Page 2, line 1: provide appropriate references for your claim suggesting that most mirrors are made of ULE,….

Response: In accordance with the reviewer's comments, the authors acknowledge that the statement "Most of the above mirrors are made of ULE, Zerodur or Borosilicate" lacked precision. It has been revised to accurately reflect that ULE and ZERODUR materials are widely used in the manufacturing of large aperture aspherical mirrors. Relevant references have been added to support this updated information. This change aims to provide a more accurate and substantiated description of the materials commonly employed in large aperture aspherical mirror fabrication.

  1. Comment: Page 2, line 3 and line 26 and line 23: Provide a better reference you’re your claim. Maybe a textbook with a page number.

Response: we have carefully reviewed our citations and have updated them where page numbers were missing. We have now ensured that each reference includes precise page numbers, providing a clearer and more direct connection to the sourced material.

  1. Comment: Line 4: Provide an appropriate reference for your claim

Response: Added a document reference[11] at line 4 to substantiate our argument.

  1. Comment: Line 10: you said, “The amount of material removal increases significantly, especially for a SiC mirror, which requires high efficiency for material removal.” It looks true for other materials, too. Why specifically for SiC?

Response: In our research, we specifically emphasize SiC due to its distinct material properties compared to other commonly used materials like ULE (Ultra-Low Expansion Glass) or ZERODUR. SiC is known for its exceptional hardness and low material removal rate. This inherent characteristic of SiC makes it more challenging to process, particularly in applications where high precision and efficiency are critical.

While other materials may also benefit from improved material removal processes, the focus on SiC is driven by its unique combination of hardness and brittleness, which substantially lowers the material removal efficiency. This necessitates the development of more efficient processing techniques to meet the stringent demands of applications like high-precision mirrors. By highlighting SiC, our intention is to draw attention to these specific challenges and the need for tailored solutions.

  1. Comment: Line 15: The reference of 11 is not appropriate there

Response: We have already removed the reference.

  1. Comment: Some dictation problems must be addressed, not limited to “previ-ous”in line 15, “os-cil-lation” in Line 25, “en-suring” in line 12, “inFig-ure3” inLine 72, and so on.

Response: Thank you for your suggestion. We have revisited our manuscript and made corrections to the mentioned issues.

  1. Comment: - In Line 51: the word of “following” should be removed. In line 129, “zi”should be replaced by zi (z sub i).

Response: Your suggestion is very helpful to us. We have revised the above problems.

  1. Comment: Line 23; the reasons you provided should be presented as bullet points. The font in Line 37 should be corrected.

Response: Thank you for your suggestion. we have provided our reasons as bullet points in page 3 Line33-37. And font in Line 37 had been corrected.

  1. Comment: Line 41: The reference [17] is wrong. The correct reference is [18].

Response: Thank you for your reminder, we have corrected the issue.

  1. Comment: Line 42: The reference [18] is wrong. The correct reference is [19].

Response: Thank you for your reminder, we have corrected the issue.

  1. Comment: The word of “efficiency” is misused in many places in this paper. For example, don’t you mean the material removal “rate” in Line 45?What do you mean by “processing efficiency” in Line 80? Or“ convergence efficiency” in line 67? And so on.

Response: Thank you for your constructive feedback and for pointing out the concerns regarding the use of the term “efficiency” in our manuscript. You are correct in observing that “rate” may be a more appropriate term in this context. The intended meaning was to refer to the speed of material removal rather than its efficiency. We have revised our paper to accurately reflect this, changing the phrase to “material removal rate.

On Line 80 – 'Processing Efficiency': Here, our intention was to describe the effectiveness and speed of the overall processing method.

Concerning Line 67 – 'Convergence Efficiency': By "convergence efficiency," we aimed to describe the rate at which our process reaches the desired result. Acknowledging that this might have been unclear, we have amended the text to provide a more precise explanation of "convergence efficiency" in page4 Line76.

  1. Comment: In line 53, don’t you think “pitch” is more appropriate than “period”?

Response: Thank you for your suggestion. We use 'period' here because the formula used in the simulation is a trigonometric function, and thus, 'period' is employed to represent the variation in its cycle.

  1. Comment: In line 149, the author refers to [22], which does not exist.

Response: Thank you for your reminder. I apologize for the mistake, and we have already corrected the relevant literature references.

  1. Comment: Line 171; Figure 6 should cited instead of “Error! Reference source…..”

Response: Thank you for your reminder. We have corrected this error and conducted a thorough review of the manuscript.

  1. Comment: Make sure the pictures are not from your two other papers published in the Journal of Light Science and Applications.

Response: Thank you for your suggestion. We would like to assure that all the images presented in this manuscript are original and have not been published in any of our previous papers, including those in the Journal of Light Science and Applications. Each image was specifically created for this study to accurately represent our current research findings.

We believe these revisions have significantly improved the manuscript and hope that it now meets the journal's standards for publication. We appreciate the opportunity to refine our work and thank you for your contributions to this process.

Zhenyu Liu

Key Laboratory of Optical System Advanced Manufacturing Technology, Chinese Academy of Sciences

[email protected]

Reviewer 3 Report

Comments and Suggestions for Authors

Dear Authors,

The manuscript unde consideration presents simulation and practical results of high-accuracy polishing of a large SiC mirror. The topic is of a high interest for the astrononmical instrumentation community and the adjacent fields and matches teh scope of Photonics journal.

I would recommend to take into account the following poitns before making a decision on acceptance for publication:

MAJOR

1. Abstract - it is stated that the L/40 surface error is achieved for this kind of mirror for the first time ever. I suppose that the claim applies onl for the combination of the material and size, as otherwise L/40 is not an unprecedented result. Still, this statement should be supported by some comparative study. See for example Michel Bougoin, Jérôme Lavenac, Alexandre Gerbert-Gaillard, Dominique Pierot, "The SiC primary mirror of the EUCLID telescope," Proc. SPIE 10562, International Conference on Space Optics — ICSO 2016, 105623Q (25 September 2017); https://doi.org/10.1117/12.2296073. The target WFE there is 9 nm RMS, i.e L/70 at 633 nm testing wavelength.

2. There are questions regarding the novelty of the entire paper in comparison with some previous publications by the same authors. Particularly, Zhang, X., Hu, H., Wang, X. et al. Challenges and strategies in high-accuracy manufacturing of the world’s largest SiC aspheric mirror. Light Sci Appl 11, 310 (2022). https://doi.org/10.1038/s41377-022-00994-3   seem to represent some key points of the manuscript and uses exactly the same experimental surface maps.

3. Figure 2 and the corresponding text  - it would be very useful to provide more details on this model. What is the exact definition of convergency rate in this case (would be better to write an explicit equation)? Does the turning speed make a difference or is there any assumption about it?

 4. Fig 4 and the corresponding text. The netire optimization procedure is unclear. Is there a limitation on the total number of different tools used? Do you presume a set of discrete tools available in pratice? What are the optimization variables and the target functions? For example, in  sec 3 there are 2 tools with 100mm and 400 mm diameters. Were these values obtained with the optimization or chosen in advance? If these are just  the available ones, what is the optimization outcome?

5. Section 2.4  - Please give definitions for the tool removal function and the dwell time vector etc. Also, E is a surface error map, so I suppose it is a matrix. How do you unwrap it into the vector form in eq 7?  The D vector represents the dwelling time, so what is the meaning of the element indicies in this case?

6. Section 2.4. - use of the convolution/ deconvolution terms is confusing (see lines 137 and 143). By the way, the convolution symbol in Eq 6 is not very common. But even more confusing is how the convolution in Eq 6 is reduced to a product of matrix and vector in Eq 7. Usually, a decrete convolution is a sum with changing index shift. 

7. Comparison between teh simulation (Table 2) and the experiment (Fig 11) looks puzzling. The actual result is L/40 while the expected RMS error is L/14. So, why the actual result is ~3 times better? How the 2 error maps are actually related?

8. It is stated that teh actual polishing took 18 months. How long wasthe "pure" machinning time? And how is this related to the theory and simulations.

MINOR.

9. There are repeating sentenses in the abstract. Please, correct.

10. Introduction - most of the telescopes mentioned in the overview have segmented mirrors, it would be useful to have more examples of large solid mirrors there.

11. Line 2 - sometimes the Maksutov coefficent  k=E*q/a, with  E-  Young modulus, q- thermal conductivity; a-CTE, is used to compare the mirror materials and proof the best choice

12. Line 29 and Eq 3 - it would be useful to know,  which notation is used there and if the indices start from 0 or 1. Perhaps a reference to Wyant's definition of the Zernike polynomials would be useful.

13. Table1- Figure3 - it is the same information given twice

14. Line 141 - It seems that the T and D vectors are mixed in the notation

15. Line 171 - missing reference

16. Figure 8 - what causes the edge effects seen in Fig 8(d) and how influental are they?

In general, th idea of separating the surface errors into the low, mid and high frequencies domains is not new. A comprehensive study of this question and development of a clear and universal optimization algorithm would be still very useful in practice. However, in this particular case it is not clear, how the theoretical, modelling and experimental parts are connected to each other. Also it is not evident how these results could be compared with the ones known from other papers. Therefore I recommend to make an extensive revision of the manuscript before re-submitting.

Author Response

Dear Reviewer,

Subject: Manuscript ID photonics-2757637, titled "Fabrication of a 4 m SiC aspheric mirror by an optimized strategy of dividing an error map"

We would like to express our sincere gratitude to you and the reviewers for the time and effort spent reviewing our manuscript. We have carefully considered the comments and suggestions provided and have made corresponding revisions to our manuscript. Below, we address each comment in detail and revisions to the manuscript had been highlighted:

  1. Comment: Abstract - it is stated that the L/40 surface error is achieved for this kind of mirror for the first time ever. I suppose that the claim applies onl for the combination of the material and size, as otherwise L/40 is not an unprecedented result. Still, this statement should be supported by some comparative study. See for example Michel Bougoin, Jérôme Lavenac, Alexandre Gerbert-Gaillard, Dominique Pierot, "The SiC primary mirror of the EUCLID telescope," Proc. SPIE 10562, International Conference on Space Optics — ICSO 2016, 105623Q (25 September 2017); https://doi.org/10.1117/12.2296073. The target WFE there is 9 nm RMS, i.e L/70 at 633 nm testing wavelength.

Response: Thank you for your insightful comments and the references provided. We have considered your feedback and acknowledge that the claim of achieving a λ/40 RMS, though significant, is not unprecedented in the broader context of mirror polishing.

Following your suggestion, we have modified the statement in our manuscript. The updated text now reads: 'A large SiC aspheric mirror with a diameter of 4 m was polished, achieving a surface error of λ/40 RMS. To the best of our knowledge, this is the first instance of such a result for a mirror with this combination of material and size.' This revision is intended to clarify that our claim of uniqueness is specifically applicable to the combination of the mirror's material and size.

Furthermore, we have included a reference to the work you mentioned (Bougoin et al., Proc. SPIE 10562, 2017) to provide a comparative context and further substantiate our claim, which has been added to the introduction.

We hope these modifications adequately address your concerns and enhance the clarity and accuracy of our manuscript.

  1. Comment: There are questions regarding the novelty of the entire paper in comparison with some previous publications by the same authors. Particularly, Zhang, X., Hu, H., Wang, X. et al. Challenges and strategies in high-accuracy manufacturing of the world’s largest SiC aspheric mirror. Light Sci Appl 11, 310 (2022). https://doi.org/10.1038/s41377-022-00994-3   seem to represent some key points of the manuscript and uses exactly the same experimental surface maps.

Response: Thank you for your comment regarding the novelty of our paper in comparison to our previous publication: Zhang, X., Hu, H., Wang, X. et al. Challenges and strategies in high-accuracy manufacturing of the world’s largest SiC aspheric mirror. Light Sci Appl 11, 310 (2022). https://doi.org/10.1038/s41377-022-00994-3.

The article you referenced primarily provides a general overview of the entire process from the manufacturing of the mirror blank to the processing and testing of the 4m SiC mirror. In contrast, our current manuscript offers a more detailed exposition of the processing strategies used during the manufacturing phase. Therefore, the same final surface error map is utilized for illustration.

We have now included citations to the aforementioned article in our manuscript and clearly delineated the distinct focus of each work. This information has been added to the introduction to provide context and demonstrate the specific contributions of our current study.

We hope this clarification addresses your concerns and illustrates the unique contributions of our paper.

  1. Comment: Figure 2 and the corresponding text  - it would be very useful to provide more details on this model. What is the exact definition of convergency rate in this case (would be better to write an explicit equation)? Does the turning speed make a difference or is there any assumption about it?

Response: Thank you for your insightful queries regarding Figure 2 and the accompanying text in our manuscript. We have taken your feedback into account and have now included an explicit equation that defines the convergence rate for clarity.

In the revised manuscript, we have elaborated on this model by providing the exact mathematical expression used to calculate the convergence rate. This addition aims to enhance the reader's understanding of the model's parameters and their impact on the simulation results.

Regarding your question about the turning speed of the tools, we have further clarified in the text that the position of the corresponding curve on the left side of the graph indicates a stronger capability of the tool to control short-period surface form errors. We have also discussed assumptions related to the turning speed and its influence on the process, providing a comprehensive explanation of the factors involved.

We hope that these additions and clarifications address your concerns and provide a more detailed understanding of the model used in our study.

  1. Comment: Fig 4 and the corresponding text. The netire optimization procedure is unclear. Is there a limitation on the total number of different tools used? Do you presume a set of discrete tools available in pratice? What are the optimization variables and the target functions? For example, in  sec 3 there are 2 tools with 100mm and 400 mm diameters. Were these values obtained with the optimization or chosen in advance? If these are just  the available ones, what is the optimization outcome?

Response: Thank you for your insightful comments regarding Fig 4 and its accompanying text. We acknowledge that our initial explanation of the optimization procedure in the manuscript was not comprehensive enough.

In the optimization process, the number and size of the tools involved can be predetermined based on the diameter of the mirror being processed and the components of its surface form error. For the 4m SiC mirror processing, we primarily selected three different diameters of polishing pads as our tools. The optimization variables are the combinations of grinding disks and their respective path planning, with the objective function being the RMS value of the surface form error. We have supplemented the manuscript with this detailed information beneath Fig 4.

Regarding Section 3, the choice of the 100mm and 400mm tools was a result of our optimization calculations. For the particular surface form error addressed, the simulation results with a 600mm tool were less favorable, leading us to opt for the 100mm and 400mm tools in our calculations.

We hope that this additional information provides a clearer understanding of our optimization process and adequately addresses your concerns.

  1. Comment: Section 2.4  - Please give definitions for the tool removal function and the dwell time vector etc. Also, E is a surface error map, so I suppose it is a matrix. How do you unwrap it into the vector form in eq 7?  The D vector represents the dwelling time, so what is the meaning of the element indicies in this case?

Response: In response to the comments in Section 2.4, the Tool Influence Function (TIF) is defined as the instantaneous material removal by a tool executing a specific motion. The 'dwell time vector' refers to the distribution of dwell times, where each element in matrix D represents the distribution of the tool's dwell time at each stationary point. We have added an explanation before Figure 6 in the manuscript and included the reference: DENG Wei-jie, ZHENG Li-gong, SHI Ya-li, et al. Dwell time algorithm based on matrix algebra and regularization method. Optics and Precision Engineering, 2007, 15(7): 1009-1015 (in Chinese).

Regarding the surface error map E mentioned by the reviewer, the actual measurement result is indeed a discrete matrix. In the process of solving for dwell time, we transform this matrix into an n×3 format of [xi, yi, zi], where [xi, yi] represents the coordinates of the ith data point, and zi indicates the residual error at that point. This transformation is described in Section 2.3. In equation 7 (now modified to equation 8), the vector 'e' actually corresponds to zi in [xi, yi, zi], representing the residual error. Therefore, in equation 8, it is referred to as the e vector to denote its physical significance as the residual error."

.Comment: Section 2.4. - use of the convolution/ deconvolution terms is confusing (see lines 137 and 143). By the way, the convolution symbol in Eq 6 is not very common. But even more confusing is how the convolution in Eq 6 is reduced to a product of matrix and vector in Eq 7. Usually, a decrete convolution is a sum with changing index shift. 

Response: In response to the concerns raised in Section 2.4, the terminology of 'convolution/deconvolution' was initially introduced by Jones RA. The use of the ** symbol as a two-dimensional convolution operator was proposed by Kim from the University of Alexandria. The transition from Equation 6 to Equation 7 is detailed in the work of Lee H, Yang M., in their article 'Dwell time algorithm for computer-controlled polishing of small axis-symmetrical aspherical lens mold', published in Optical Engineering, 2001; 40(9):1936–1943. These references have been added to our manuscript for further clarification.

  1. Comment: Comparison between teh simulation (Table 2) and the experiment (Fig 11) looks puzzling. The actual result is λ/40 while the expected RMS error is λ/14. So, why the actual result is ~3 times better? How the 2 error maps are actually related?

Response: Thank you for your insightful comments regarding the comparison between the simulation results in Table 2 and the experimental outcomes depicted in Figure 11. I would like to clarify the observed discrepancies as follows:

1.The expected root-mean-square (RMS) error of λ/14, as presented in Table 2, represents the outcome of the current round of simulation for the manufacturing process.

2.Figure 10 illustrates the actual results from this round of experimental manufacturing.

3.The λ/40 result shown in Figure 11 corresponds to the final outcome of the 4m SiC mirror after the completion of the manufacturing process.

To address your concerns and avoid any potential confusion, I have revised Figure 11. The updated figure now includes a photograph of the actual SiC mirror upon the completion of its manufacturing. This should provide a clearer understanding of the relationship between the simulation and experimental results and explain why the actual result is approximately three times better than the initially expected RMS error.

I hope this explanation resolves the issues raised and I appreciate your attention to the details of our work.

  1. Comment: It is stated that teh actual polishing took 18 months. How long wasthe "pure" machinning time? And how is this related to the theory and simulations.

Response: Thank you for your question regarding the duration of the actual polishing process described in our manuscript. I appreciate the opportunity to provide further clarity on this aspect.

The actual polishing of the component, as mentioned, took 18 months. This duration includes not only the time spent directly in polishing but also encompasses preparatory and intermittent processes such as setup, inspection, and adjustments based on ongoing results.

The total time spent on processing and measurement activities was 377 days, with the measurement phase consuming over half of this duration. The significant amount of time devoted to measurement can be attributed to the challenges associated with obtaining stable and trustworthy measurement data for high-precision, large-aperture (4m class) aspheric surfaces. The 'pure machining time', as mentioned by the reviewer, was approximately 180 days.

The simulations predicted a required polishing time of 120 days, which was based on factors like material removal rate, Surface error distribution. The difference between the simulated and actual "pure" machining times can be attributed to unforeseen material behavior, tool wear rates, edge effect control. This comparison is crucial as it provides insights into the accuracy and applicability of our theoretical models and simulation tools in practical, real-world scenarios.

  1. Comment: There are repeating sentenses in the abstract. Please, correct

Response: Thak you for your suggestion. In response to your observation, we have thoroughly rewritten the abstract. This revised version eliminates the repetitive statements and enhances the overall clarity and succinctness.

  1. Comment: Introduction - most of the telescopes mentioned in the overview have segmented mirrors, it would be useful to have more examples of large solid mirrors there.

Response: Thank you for your suggestion to include more examples of large solid mirrors in the introduction. In response to your valuable input, we have updated the introduction to incorporate additional references related to the Large Binocular Telescope (LBT), the Subaru Telescope, and the Very Large Telescope (VLT). These additions provide a more comprehensive overview of large solid mirror telescopes, thereby enhancing the context and relevance of our study in this field.

  1. Comment: Line 2 - sometimes the Maksutov coefficent  k=E*q/a, with  E-  Young modulus, q- thermal conductivity; a-CTE, is used to compare the mirror materials and proof the best choice

Response: Thank you for your insightful suggestion. In response to your recommendation, we have revised the relevant section of our paper to incorporate a more detailed discussion on this aspect. Additionally, we have provided a specific literature reference to substantiate our description of the SiC (Silicon Carbide) mirror material properties. The reference is as follows:

Xuejun Zhang, "Manufacturing and testing large SiC mirrors in an efficient way", Proc. SPIE 9628, Optical Systems Design 2015: Optical Fabrication, Testing, and Metrology V, 96280S (24 September 2015).

  1. Comment: Line 29 and Eq 3 - it would be useful to know,  which notation is used there and if the indices start from 0 or 1. Perhaps a reference to Wyant's definition of the Zernike polynomials would be useful.

Response: Thank you for your inquiry regarding the notation and indexing used in Line 29 and Equation 3 of our manuscript. In our work, we have employed the "Fringe" indexing scheme, which starts at 0 instead of 1 (subtract 1). This method is widely utilized, including in interferogram analysis software for Zygo interferometers and the open source software DFT Fringe.

Additionally, we have now included a reference to James C. Wyant's work immediately following Equation 4 in our paper. This reference provides a clearer theoretical framework and aligns our methodology with established practices in the field.

  1. Comment: Table1- Figure3 - it is the same information given twice

Response: Thank you for your observation regarding the apparent duplication of information between Table 1 and Figure 3 in our manuscript. We acknowledge your concern and would like to clarify the purpose of this presentation format.

Figure 3 is designed to provide a more intuitive visual representation of the trends in 'Dwell time and convergence rate as a function of tool sizes'. This visual approach aids in quickly grasping the overall trends and patterns. On the other hand, Table 1 presents the exact data in a detailed format. This allows interested readers to access precise information and facilitates a deeper understanding of the specific data points.

We believe that this dual approach effectively caters to different reader preferences, offering both a quick visual summary and detailed data for in-depth analysis.

  1. Comment: Line 141 - It seems that the T and D vectors are mixed in the notation

Response: Thank you for pointing out the inconsistency in the notation of the T and D vectors on Line 141 of our manuscript. We apologize for this error.

We have carefully reviewed the section and corrected the notation to accurately reflect the intended vectors. We appreciate your attention to detail, which has helped us improve the precision and clarity of our manuscript.

  1. Comment: Line 171 - missing reference

Response: Thank you for pointing out the missing reference on Line 171 of our manuscript. We have reviewed this section and added the appropriate reference to rectify this oversight.

  1. Comment: Figure 8 - what causes the edge effects seen in Fig 8(d) and how influental are they?

Response: Thank you for your inquiry about the edge effects observed in Figure 8(d). During the mirror polishing process, when the tool center extends beyond the mirror's edge, it can lead to a flipping action. Thus, in our path planning, the dwell points are required to maintain a certain distance from the edge. This results in the dwell time solving process being a non-complete convolution, leading to errors at the edges. This issue affects the convergence efficiency of the mirror surface shape and is an area for further research in our work.

We believe these revisions have significantly improved the manuscript and hope that it now meets the journal's standards for publication. We appreciate the opportunity to refine our work and thank you for your contributions to this process.

Zhenyu Liu

Key Laboratory of Optical System Advanced Manufacturing Technology, Chinese Academy of Sciences

[email protected]

Round 2

Reviewer 2 Report

Comments and Suggestions for Authors

The author did all the modifications I suggested. This is a good paper, and I'd recommend it for publication. 

Author Response

Dear Reviewer,

First and foremost, I would like to express my sincere gratitude for your valuable comments and suggestions during the review process. Your expertise has been instrumental in enhancing the quality of my paper.

Following your initial review comments, I have made the necessary modifications and additions to the manuscript. I have endeavored to ensure that each amendment accurately reflects your suggestions, while maintaining the original intent and integrity of the research.

I am pleased to learn that the revised paper has met with your approval and is recommended for publication. I believe that this paper offers new insights in my field of study and could serve as a valuable reference for future research.

Thank you once again for your time and expert advice.

Sincerely

Zhenyu Liu

Key Laboratory of Optical System Advanced Manufacturing Technology, Chinese Academy of Sciences

[email protected]

Reviewer 3 Report

Comments and Suggestions for Authors

Dear Authors, 

thank you for taking into account my remarks and preparing the revised version of the manuscript. All of my points have been addressed and I believe that the manuscript is clearer for a reader now. As an optional final modification I would yet recommend to inclue more detail, which you provided regarding the polishing  time estimation, into the main text. In your reply it is stated that the time predicted by the simulations is about 120d, while the actual polishing duration was 180d. It is useful to show that the model does not allow one to predict the exact value, but it's yet precise enough to compare different polishing strategies. Perhaps, you could also show how the time estimations for separate tools (in hours), given in Tables 1 and 2 correspond to the total processing time in days.

Otherwise , the paper can be accepted for publication.

Author Response

Dear Reviewer,

Thank you once again for your constructive comments and for acknowledging the improvements made in the revised version of the manuscript. I greatly appreciate your suggestion to include more detailed information about the polishing time estimation in the main text.

In response to your recommendation, I have added additional details at the end of Section 3 of the manuscript. This includes a more comprehensive analysis of the actual processing time, as well as an in-depth comparison of the impact of using individual tools versus a combination of tools on the total processing time. These new sections aim to provide a clearer understanding of the discrepancies between the simulated and actual polishing durations.

I hope these additions satisfactorily address your recommendations and further enhance the manuscript's clarity and utility for readers. Your feedback has been invaluable in refining this paper, and I am grateful for your guidance throughout this process.

Thank you for your continued support and insightful advice.

Sincerely,

Zhenyu Liu

Key Laboratory of Optical System Advanced Manufacturing Technology, Chinese Academy of Sciences

[email protected]